# Interaction of Cytochrome C Oxidase with Steroid Hormones

**DOI:** 10.3390/cells9102211

**Published:** 2020-09-29

**Authors:** Ilya P. Oleynikov, Natalia V. Azarkina, Tatiana V. Vygodina, Alexander A. Konstantinov

**Affiliations:** A.N. Belozersky Institute of Physico-Chemical Biology, M.V. Lomonosov Moscow State University, Leninskie Gory 1, Bld. 40, 119 992 Moscow, Russia; oleynikov.biophys@gmail.com (I.P.O.); vygodina@belozersky.msu.ru (T.V.V.); konst@gmail.com (A.A.K.)

**Keywords:** cytochrome oxidase, steroid hormones, regulation

## Abstract

Estradiol, testosterone and other steroid hormones inhibit cytochrome *c* oxidase (CcO) purified from bovine heart. The inhibition is strongly dependent on concentration of dodecyl-maltoside (DM) in the assay. The plots of K_i_ vs [DM] are linear for both estradiol and testosterone which may indicate an 1:1 stoichiometry competition between the hormones and the detergent. Binding of estradiol, but not of testosterone, brings about spectral shift of the oxidized CcO consistent with an effect on heme *a_3_*^3+^. We presume that the hormones bind to CcO at the bile acid binding site described by Ferguson-Miller and collaborators. Estradiol is shown to inhibit intraprotein electron transfer between hemes *a* and *a_3_*. Notably, neither estradiol nor testosterone suppresses the peroxidase activity of CcO. Such a specific mode of action indicates that inhibition of CcO activity by the hormones is associated with impairing proton transfer via the K-proton channel.

## 1. Introduction

Cytochrome *c* oxidase (CcO) is a key enzyme of aerobic metabolism and of oxidative phosphorylation, in particular, providing living organisms with access to the usage of oxygen reduction energy (reviewed, [1,2,3]). It is conceivable that an enzyme of such high importance should be subject to thorough control at all levels of cell metabolism—biosynthesis, assembly and regulation of the assembled enzyme turnover in response to various intracellular stimuli, for example by signal molecules emerging in the cells. The latter type of regulation is particularly suitable for in vitro biochemical studies. Regulation of CcO activity by nucleotides (see numerous works of B. Kadenbach and his laboratory, for instance [4]) and gaseous ligands of the oxygen-reducing center (see the review by Cooper & Brown [5]) has been amply studied. Recently, direct modulation of the mitochondrial CcO activity by Ca^2+^ and Na^+^ ions binding to a special cation-binding center was described (see the series of works from our laboratory, for example [6,7,8,9,10,11]).

A promising new venue for research of CcO regulation has been provided by recent works of Ferguson-Miller and collaborators. In a series of papers [12,13,14,15,16], the Michigan group described a conserved bile acid binding site (BABS) in the crystal structure of cytochrome *c* oxidase (CcO) from *Rhodobacter sphaeroides* or bovine heart mitochondria. The site is located near the inner side of the membrane close to the entrance of the so-called K-proton channel and binds a variety of amphiphilic ligands of diverse nature sharing structural similarity, such as bile acids, thyroid hormones, retinoic acid and many other compounds; see Buhrow et al. [15]. Binding of the ligands was found to bring about inhibition or sometimes stimulation of the enzyme activity. Structural considerations allowed to rank steroid hormones, such as testosterone or estradiol, high in the list of potential ligands of the BABS [15]. Surprisingly, no effect of steroid hormones on the activity of CcO from *R. sphaeroides* was found, whereas many other ligands of the site, e.g., thyroid hormone T3, inhibited the enzyme.

We surmised that CcO from animal mitochondria may be a more appropriate object for the studies of hormone action than the bacterial enzyme. In this work we show that sex hormones, testosterone and estradiol, as well as several other steroid hormones can markedly inhibit activity of CcO purified from bovine heart mitochondria. The inhibition is accompanied by deceleration of electron transfer between hemes *a* and *a_3_*. Interestingly, peroxidase activity of the enzyme is not affected by the steroid hormones, supporting the proposal by Hiser et al. [14] that the inhibition of the oxidase reaction by the ligands of BABS is caused by their action on proton delivery to the oxygen-reducing site via the K-proton channel.

## 2. Materials and Methods

### Chemicals

Testosterone, estradiol-3-benzoate, sodium dithionite, cytochrome c (type III from equine heart), TMPD (N,N,N’,N’-tetramethyl-p-phenylenediamine), l-ascorbic acid, potassium ferricyanide, catalase and hydrogen peroxide were from Sigma-Aldrich (Saint Louis, MO, USA), hexaammineruthenium (III) was from Alfa Division (Ward Will, MA, USA). Hydrogen peroxide solution (about 30%) was kept at 4 °C and its concentration was checked spectrophotometrically using molar extinction coefficient ε_240_ = 40 M^−1^⋅cm^−1^ [17]. Dodecyl-maltoside of “Sol-Grade” type was purchased from Anatrace (Maumee, OH, USA). pH-buffers and EDTA (ethylenediaminetetraacetic acid) were from Amresco (Radnor, PA, USA). o-dianisidine dihydrochloride was from ICN Biomedicals Inc (Irvine, CA, USA). Testosterone was dissolved in ethanol and estradiol-3-benzoate in dimethyl sulfoxide for stock solutions.

*Cytochrome oxidase* was isolated from bovine heart mitochondria using the modified method of Fowler et al. [18] as described previously by Vygodina et al. [8]. Concentration of the enzyme was determined from the difference absorption spectra (dithionite reduced minus air oxidized) using molar extinction coefficient ε_605-630_ = 27 mM^−1^⋅cm^−1^.

*Cytochrome oxidase activity assay*. Oxygen uptake rates were measured with a covered Clark-type electrode by an Oxytherm instrument (Hansatech, UK) in a thermostatted cell at 25 °C with permanent stirring. The assays were performed in a basic medium containing 50 mM Hepes, 50 mM Tris, pH 7.5 and 0.1 mM EDTA. The medium was also supplemented with indicated concentrations of DM which are given throughout the text in mM (0.05% ≈ 1.0 mM DM). 5 mM ascorbate, 0.1 mM TMPD and 10 µM cytochrome *c* were used as the oxidation substrate. Other details are indicated in the legends to figures.

*Peroxidase activity of CcO* was assayed as described by Vygodina and Konstantinov [19] by following spectrophotometrically peroxidation of 0.2 mM *o*-dianisidine at 432 nm minus 580 nm in a dual-wavelength SLM Aminco DW-2000 spectrophotometer (USA). The reaction mixture contained 0.6 µM CcO in the aerobic basic medium supplemented with different concentrations of DM. No *o*-dianisidine oxidase activity was observed and the peroxidase reaction was initiated by addition of 4 mM H_2_O_2_.

*Absorption spectra of CcO* were recorded with a Cary 300 Bio spectrophotometer (Varian, Palo Alto, CA, USA) in semi-micro cuvettes with blackened walls and 10 mm light pathway (Hellma, Müllheim, Germany), in the basic medium at pH 8.0 supplemented with indicated different additions. The kinetics of spectral changes was monitored in a dual-wavelength mode in a SLM Aminco DW-2000 spectrophotometer.

*The kinetics of CcO reduction* by dithionite in the presence of ruthenium hexammine (RuAm) was studied in an Applied Photophysics SX-20 stopped-flow spectrophotometer (UK) operated in a diode array mode, using a 20 µL cell with 1 cm optical pathway. The spectra in the 280–720 nm range were collected with a minimal interval of 1 ms. The assay medium contained 100 mM Hepes, 100 mM Tris, 50 mM KCl, 100 µM EGTA, catalase (2 µL/5 mL of the solution, indicated activity 23,000 units/mg protein), 0.05% DM, pH 8.0. Aerobic buffer with 6 µM CcO and 0.4 mM or 1 mM estradiol was rapidly mixed with the equal volume of the same buffer containing 40 mM sodium dithionite and 10 mM RuAm.

*Procession of the stopped-flow data* was performed with ProKineticist software provided with the Applied Photophysics SX-20 instrument, as well as by using Origin 7 and Origin 9 Microcal software.

## 3. Results

### 3.1. Inhibition of CcO Oxidase Activity by Steroid Hormones

At variance with Buhrow et al. [15], we found that several steroid hormones bind to the isolated CcO bringing about considerable inhibition of the enzyme activity.

#### 3.1.1. Inhibition of CcO Oxidase Activity by Estradiol

Figure 1 shows the inhibitory effect of estradiol on oxygen consumption by isolated bovine CcO oxidizing ascorbate + TMPD in the presence of cytochrome *c*. The effect of estradiol depends significantly on the concentration of DM in the assay. Panel A gives representative oxygraph recordings. At 1 mM DM, 2 mM estradiol brings about near 4-fold inhibition of the activity (trace *1*). Notably, the inhibition is not instantaneous but takes a couple of minutes to develop. In the presence of 20 mM DM (trace *2*), the same addition of estradiol results in only a slight inhibition. Moreover, the inhibition induced by 1 mM estradiol at 1 mM of DM can be partially released by subsequent addition of high concentration of the detergent (trace *3*).

Concentration dependence of the estradiol-induced inhibition was measured at different concentrations of DM in the assay (Figure 1B). The titrations fit to hyperbolic curves tending to 100% inhibition at infinite concentration of the inhibitor. It can be seen that the higher the concentration of DM, the more estradiol is required to inhibit the enzyme. Notably, a distinct lag phase is observed at high concentrations of DM (see traces *3–5*).

The normalized activity, *v*, in the presence of estradiol, *I*, can be described by hyperbolic equation shifted on the abscissa scale by a value of lag phase, *L*:(1)$$v=\frac{1}{1+\frac{ [I]-L}{K_{i\left(app\right)}}}$$
where *K_i(app)_* is the apparent value of inhibition constant in the presence of a given DM concentration.

We propose that the lag phase may reflect binding of estradiol with the empty DM micelles. As one can see from the Inset in Figure 1B, the lag-phase value depends on DM concentration almost linearly in the range of 2–10 mM and then levels off.

The plot of K_i(app)_ vs [DM] is shown in Figure 1C. The dependence is close to linear in the entire range of DM concentrations studied. This points out a 1:1 competition between estradiol and DM for binding with the enzyme, with the slope tangent equal to 0.25 which is the ratio of the respective affinities. Extrapolation of the plot to zero concentration of DM yields true K_i_ value of 0.37 mM. The estimated dissociation constant for the competitor, DM, in the absence of estradiol is K_c_ = 1.47 mM.

#### 3.1.2. Testosterone Induced Inhibition of CcO Oxidase

Inhibition of cytochrome oxidase activity by testosterone is shown in Figure 2. Under certain conditions (high concentrations of both testosterone and DM, e.g., Figure 2A, trace *1*) the inhibition directly observed can be as high as 3-fold and occurs instantaneously. As with estradiol, the inhibitory effect of testosterone depends on DM concentration, but in a more complex way. Addition of excess testosterone (4 mM) at low concentration of DM (1 mM) inhibits the activity but weakly, whereas strong inhibition is induced by this concentration of the hormone in the presence of 20 mM DM (traces *2* and *1*, respectively). Accordingly, when DM concentration is raised in the course of respiration (trace *3*), the inhibition by the initially present excess testosterone (3 mM) is not released as it was observed with estradiol but rather is slightly augmented.

The complex pattern of CcO inhibition by testosterone can be better understood from the titrations made at different concentrations of DM (Figure 2B). At low [DM], the activity does not tend to zero with increased concentrations of testosterone as in the case of estradiol, but rather the titration curves level off at some saturation levels attained at relatively low concentrations of the hormone (cf. the dashed lines). The maximal level of inhibition by excess testosterone grows with increased concentrations of DM in the assay from ca. 25% at 1 mM of the detergent to 70–85% at 20–40 mM of DM. At high concentrations of DM, the testosterone titration curves reveal a clear lag phase (see the initial parts of the titration curves *2*–*5*) and as with estradiol, we assign this lag phase to hormone binding with the empty DM micelles.

In order to determine K_i(app)_, the titration curves in Figure 2B were approximated by Equation (2) that connects normalized activity, *v*, with testosterone concentration, [*I*]. It is similar to Equation (1) but has an additional parameter *R* which denotes the fraction of the activity resistant to inhibition:(2)$$v=\frac{1-R}{1+\frac{ [I]-L}{K_{i\left(app\right)}}}+R$$

Equation (2) fits well the experimental data after the end of the lag-phase (see other details in the legend). All the parameters (K_i(app)_, *L* and *R*) were found to depend on DM concentration. In particular, the dependence of the lag phase on [DM] is linear throughout the entire concentration range explored, i.e., 1–40 mM (see the Inset in Figure 2B, circles, left Y axis). Dependence of the extent of maximal inhibition (which is 1-R) on [DM] is also shown in the Inset (triangles, right Y axis). As seen, the maximal inhibition grows from ~25% to ~90%.

The dependence of K_i(app)_ on [DM] is shown in Figure 2C. It points to a competition between testosterone and DM with the 1:1 stoichiometry. Although the scatter of the experimental points is quite large here (due to low solubility of testosterone), we assume a linear dependence, as in all other cases; see below Table 1 in Section 4. The slope tangent gives the ratio K_i_:K_c_ = 0.05. Extrapolation to zero DM concentration yields the true K_i_ value for testosterone as low as 80 µM, whereas the K_c_ value is now determined as 1.3 mM (compare to 1.47 mM as estimated in Figure 1C).

### 3.2. Effect of Estradiol on CcO Spectral Properties

#### 3.2.1. Estradiol Shifts Absorption Band of Heme *a_3_*^3+^

We found that binding of estradiol not only inhibits the enzyme but also perturbs the absorption spectrum of the oxidized CcO. No effect of estradiol on the absorption spectrum of the reduced CcO was observed. Some typical data are shown in Figure 3. Small but distinct changes are observed at estradiol concentrations as low as 0.5 mM (Figure 3A, spectra *1–4*) but much better spectra are obtained at 1 mM of the hormone (spectra *5–7* in Figure 3A). Note that at 13 mM of DM as used in this experiment, the indicated concentrations of estradiol correspond to ca. 0.14 and 0.29 of the K_i(app)_ value obtained in the activity inhibition studies (see Figure 1C). The symmetric S-shaped difference spectrum induced by estradiol (Figure 3A, *5–7*, Figure 3B, *1*) with a maximum at 437 nm and a trough at 415 nm is typical of a red shift of ferric heme *a_3_* Soret band induced by strong ligands such as cyanide. The magnitude of 10 mM^−1^⋅cm^−1^ obtained at 1 mM of the hormone corresponds to ca. 1/5 of the effect induced by cyanide, see van Buuren et al. [20]. In this case the spectral shift develops rapidly enough, reaching completion in 3–5 min after the hormone addition (Figure 3A,C, trace *2*) in reasonable agreement with the time course of the inhibitory action of estradiol that takes a few minutes to develop (Figure 1A).

#### 3.2.2. Cyanide Interferes with Estradiol-Induced Spectral Effect in CcO

The spectral perturbation of CcO induced by estradiol is strongly modified by cyanide (Figure 3B). The S-shaped difference spectrum typical of heme *a_3_*^3+^ red shift is no longer observed and is replaced by a trough at 428 nm corresponding to the maximum of *a_3_*^3+^-CN complex, which develops very slowly (for at least 3 h, Figure 3B, spectra *2*, *3,*
Figure 3C, trace *3*).

These data allow to suggest that the spectral perturbation of the oxidized ligand-free CcO induced by estradiol is associated with heme *a_3_*. Perturbation of the heme *a_3_* spectrum would be consistent with the proposal by Hiser et al. [14] that the steroid hormones and other ligands of the BABS affect the K-proton channel interacting with the oxygen-reducing binuclear site of CcO.

### 3.3. Mechanism of CcO Inhibition by Steroid Hormones

To the best of our knowledge, mechanism of the inhibitory action of the BABS ligands on CcO activity has not been studied experimentally so far.

#### 3.3.1. Inhibition of Electron Transfer between Hemes *a* and *a_3_* by Estradiol

We investigated effect of estradiol on the kinetics of electron transfer from heme *a* to heme *a_3_*. To this end, the kinetics of the hemes’ reduction by dithionite in the presence of RuAm was studied. In the experiment shown in Figure 4, 3 µM CcO was mixed rapidly with 20 mM dithionite and 5 mM RuAm and the reaction time course was followed by taking the spectra each ms in the 280–720 nm range in a rapid mixing diode array spectrophotometer. Under these conditions, heme *a* is almost fully reduced within the mixing time as evidenced by the spectra in the α-band region (panel A, *visible*), which is followed by time-resolved electron transfer to heme *a_3_* observed in the Soret region (panel A, *Soret*) [6,21].

As shown in Figure 4B, the kinetics of heme *a_3_* reduction monitored at 445 nm vs the 407 nm reference is noticeably decelerated in the presence of 200 µM and, especially, 500 µM estradiol (compare control trace *1* with traces *2* and *3*, respectively). The effect was reproduced with several samples.

Kinetic analysis of the traces shown in Figure 4B revealed two phases which upon the conditions used (see above) reflect reduction of heme *a_3_*. In the control (trace *1*), the rate constants are 105.3 s^−1^ and 6.15 s^−1^, which is in agreement with our earlier data at these concentrations of dithionite and RuAm [6]. In the presence of 0.2 mM estradiol (trace *2*) the rate constants change to 67.75 s^−1^ and 5.3 s^−1^. An increase in the estradiol concentration to 0.5 mM (trace *3*) yields a further decrease of the values to 39.6 s^−1^ and 2.5 s^−1^, respectively. The contributions of the rapid and slow phases make up about 70% vs 30% in all three cases. These estimates are consistent with the data obtained in the experiments with an oxygen electrode. For example, according to the rapid mixing data, 0.5 mM estradiol decreases by ca. 60% the rate constants of the both phases (without a notable effect on their contributions), and the same concentration of estradiol inhibits oxygen consumption approximately by half (Figure 1B, curve *1*; Figure 1C). Unfortunately, higher concentration of estradiol could not be used in the optical experiments because of increased sample turbidity.

#### 3.3.2. Effect of the Hormones on Peroxidase Activity of CcO

Taking into consideration close proximity of the BABS to the entry point of the K-proton channel, it was speculated that binding of the ligands to the amphipathic site could result in impairment of proton transfer to the binuclear site via the K-pathway, see Hiser et al. [14]. Reduction of dioxygen to two water molecules by cytochrome oxidase occurs in two sequential phases. In the first phase denoted as *eu-oxidase* (see Konstantinov [22]), O_2_ bound to the reduced binuclear site withdraws 4 electrons from CcO, the O-O bond is cleaved, and the first H_2_O molecule is formed, which requires uptake of 2 “chemical” protons delivered via the K-channel. The second oxygen atom of the dioxygen remains as the oxene ligand bound to iron of ferryl heme *a_3_* in a Fe^4+^=O^2^^−^ /Tyr^∙^ compound. This intermediate denoted as FI-607 (or P_m_ in the outdated terminology) is two oxidizing equivalents deficient relative to the resting oxidized state of CcO and is homologous to Compound I of peroxidases. The second oxygen atom is subsequently reduced to H_2_O by two sequential electron transfers from exogenous substrate, a process fully analogous to reduction of Compound I to the resting state in peroxidases. Formation of this second water molecule by CcO was denoted therefore as the *peroxidase* phase of the CcO catalytic cycle. The protons consumed in H_2_O formation in the peroxidase phase are delivered exclusively via the D-proton channel and the K-channel is not involved in this part of the CcO catalytic cycle.

The binuclear site of CcO can use H_2_O_2_ instead of O_2_ as the electron acceptor [19,22]. In this case, the eu-oxidase phase of the catalytic cycle can be bypassed and the enzyme cycles exclusively through the peroxidase phase. Accordingly, the peroxidase activity of CcO is blocked by mutations in the D-channel whereas mutations in the K-channel such as K362M replacement in CcO from *R. sphaeroides* do not affect the peroxidase activity of CcO while fully inhibiting the oxidase activity of the enzyme, see Vygodina et al. [23]. Therefore, we were excited to see that testosterone and estradiol, while suppressing the oxidase reaction, do not inhibit peroxidase activity of CcO (Figure 5). This observation suggests that the inhibitory effect of the steroid hormones on the activity of CcO may be indeed associated with impairment of proton transfer via the K-channel.

## 4. Discussion

Our work shows that testosterone and estradiol can inhibit significantly isolated cytochrome oxidase from bovine heart mitochondria. The inhibition of the isolated bovine CcO has been observed also with a number of other steroid hormones including progesterone and DHEA, as well as with vitamin D which is classified as secosteroid (Table 1).

All the tested steroid compounds induced inhibition of CcO activity at 10^−5^–10^−4^ M concentration. All of them revealed competition with dodecyl-maltoside. None of the assayed compounds inhibited peroxidase activity of CcO, with the exception for cholecalciferol which, however, affects peroxidase reaction in concentrations an order of magnitude higher than oxidase activity (Table 1).

These findings corroborate the proposal of Ferguson-Miller and collaborators based on structure considerations that steroid hormones are to be highly ranked in the list of potential ligands of the bile acid binding site of CcO. At the same time, experiments of the East Lancing group did not reveal inhibition of CcO by testosterone, estradiol or other steroid hormones (cf. Supplemental Figure 2 to Buhrow et al. [15]), while many other amphipathic ligands of the BABS, e.g., retinoic acid, or thyroid hormone T3, strongly inhibited the enzyme [15].

The discrepancy between the results of the present work and that of Buhrow et al. [15] could be explained by different experimental conditions. Firstly, the CcO species used in the two studies are different. Ferguson-Miller and collaborators worked with CcO from *R. sphaeroides* whereas bovine heart oxidase was studied in our experiments. It would be tempting to speculate that the animal CcO can be more susceptible to inhibition by animal hormones than the bacterial enzyme, emphasizing physiological relevance of the effect. However, discrepancy between the data here and in [15] may probably be explained by some other significant differences in the experimental conditions. For instance, the activity assay conditions in [15] ([DM] = 0.2 mM) were not optimal for the inhibitory effect of testosterone to be revealed. With the concentrations of testosterone (100 µM) and DM (1 mM) close to those used in [15], we observed inhibition of bovine oxidase by only 13% (see Figure 2B, curve *1*). In order to clarify the issue, it would be best to carry out side-by-side experiments with the bovine and bacterial oxidases under identical conditions. Our preliminary experiments reveal inhibition of *R. sphaeroides* CcO by estradiol.

While Ferguson-Miller and collaborators described inhibition of CcO by a number of amphipathic ligands of the BABS, the mechanism of the inhibition was not investigated earlier. Ferguson-Miller and collaborators [14] pointed out close proximity of the BABS to the entry of the K-proton channel and proposed that the amphipathic ligands may impair proton delivery to the oxygen-reducing site via the K-pathway. This proposal was supported by strong modulation of the BABS ligand effects on the activity of CcO by mutations in residue E101 in Subunit II that is thought to be the entry point for protons in the K-pathway, see Hiser et al. [16]. The inhibition of intra-molecular electron transfer from heme *a* to heme *a_3_* by estradiol observed in our work is consistent with inhibition of the K-channel. A distinctive feature of CcO inhibition induced by blocking the K-channel consists in that the peroxidase activity of the enzyme is not affected (see Vygodina et al. [23]) because the K-channel is not involved in proton delivery to the binuclear site in the so-called peroxidase phase of the catalytic cycle [22,23]. As shown in this work, testosterone, estradiol and other steroids, while suppressing the oxidase reaction, actually do not inhibit peroxidase activity of CcO (Figure 5, Table 1). This observation strongly supports the proposal that the inhibitory effect of the steroid hormones on the activity of CcO may indeed be associated with impairment of proton transfer via the K-channel. Effect of estradiol on the proton conducting relay of the K-channel is also in agreement with the spectral shift of heme *a_3_* induced by the hormone (Figure 3). Indeed, the K-channel delivers protons right to the binuclear site and perturbation of the proton relay could affect H_2_O/OH^−^ equilibration at the 6-th axial position of heme *a_3_*, leading to partial high-to-low-spin transition of the ferric heme.

An obvious important question is whether the inhibition of the isolated CcO by steroid hormones may be relevant to physiology. The first point to be concerned is the hormone concentration range. Physiological concentration of the steroid hormones in the blood serum is around 10^−7^–10^−6^ M for the total [24], ca. 1 × 10^−10^–1.2 × 10^−8^ M for free testosterone and ca. 2x10^−8^–5x10^−7^ M for free estradiol [25,26] that seems to be incompatible with the K_i_ values of 10^−5^–10^−4^ M determined for the hormone action on CcO in our experiments. However, it is noted that steroid hormones are very lipophilic and partition from water to octanol with a coefficient of 10^3^–10^4^ [27], so that their concentration in the mitochondrial membranes may be very much higher than in blood serum. It is widely accepted that accumulation of hormones in membranes can be an important step of their action mechanism in the cell, see Mayne et al. [28]. Thus, the highest partition coefficient of 10^4^ brings the values of the order of 10^−8^ M (the free hormones in the blood serum) to the hormone concentrations in the mitochondrial membranes close to the estimated K_i_ (ca. 10^−4^ M). Therefore, quite noticeable inhibition of cytochrome oxidase by physiological concentrations of the steroid hormones looks possible.

At the same time, our data may be relevant to the experimental studies of steroid hormones action in vivo, studies with whole animals or (and particularly so) with cell cultures. Concentration of the hormones employed in such studies often reach a submillimolar or even millimolar range [29,30,31,32,33]. For example, in the paper of D’Ascenzo et al. [29], the following IC_50%_ values are cited for inhibition of cell growth by anabolic steroides: testosterone, 100 µM; androstenedione, 375 µM; nandrolone, 9 µM; norandrostenedione, 500 µM; norandrostenediol, 6 mM. Effective concentrations of tens to hundreds of micromoles are also given in the rest of the works mentioned above.

The second point as to the physiological relevance of our data is that inhibition of the isolated CcO by steroid hormones has been observed under conditions rather far from physiological. For instance, all the data have been obtained in the presence of a detergent dodecyl-maltoside and it remains to be established whether the inhibition by the hormones can take place in case of the membrane-bound enzyme. Experiments with intact mitochondria and liposome-reconstituted CcO are required in order to resolve this question.

Finally, we would like to discuss the role of DM in the inhibition of CcO by steroids. Experiments with both testosterone and estradiol suggest that DM competes with the steroid hormones for binding with the enzyme. Notably, the close values of affinity between DM and CcO are determined independently in the two cases of inhibition (K_c_ = 1.47 mM and 1.3 mM, see Figure 1C and Figure 2C, respectively). Moreover, we obtained nearly the same value (1.2 mM) in our study on the inhibition of CcO by Triton X-100, the detergent which structurally resembles steroids in some respects (the paper is submitted for publication). One can assume from these data that DM specifically interacts with the enzyme at the site with high affinity to some native ligand. This conclusion is in agreement with the structure analyses in [12,14], that visualize decyl-maltoside molecule binding very close to the BABS with a hydrophobic tail binding to a groove near the amphipathic region to which the steroid group of the bile acids adheres, and with the maltoside head being able to reach the steroid group binding motif of the amphiphilic site competing with the typical BABS-ligand for binding with CcO.

How can it be that DM at the same time increases the maximal inhibition obtained with excess testosterone? At this time we can only speculate on this issue. One possible explanation is that there are two DM binding sites involved. Binding at one site confers CcO ability to bind testosterone, presumably stabilizing a putative testosterone-reactive conformational state of the enzyme, and so increases the maximal level of inhibition that can be attained with saturating concentrations of the hormone. DM bound at the other site competes with testosterone for binding with the BABS which explains the growth of K_i_ with increase in [DM]. These two putative binding sites may bind two different molecules of DM, however, binding of a single molecule of DM can fit to this purpose as well. Namely, binding of the hydrophobic tail to the groove could be responsible for stabilizing the testosterone-reactive state of CcO, whereas overlapping of the maltose head group with the steroid-binding motif could hamper testosterone binding to the BABS.

In any case, the different character of the influence of DM on the inhibition imposed by estradiol and testosterone points to the probable difference in mutual arrangement of the hormone and detergent molecule in these two cases. It is possible that the absence of spectral changes in CcO incubated with testosterone (as opposed to estradiol, see Table 1) reflects the shifted disposition of BABS ligands which causes a difference in their sum effect on the K-channel structure and, ultimately, on the binuclear center.

It is worth noting that DM can be viewed as a structural analog of phospholipids, see Qin et al. [34]. So, the effects of DM on the hormone binding with CcO may in fact mimic the effects imposed in vivo by endogenous phospholipids binding to a site adjacent to the BABS and controlling reactivity of the BABS to amphipathic ligands.

In conclusion, we presume in agreement with Ferguson-Miller and collaborators [12] that CcO is endowed with a specific regulatory site, capable of binding various amphiphilic ligands, including steroid hormones. Binding of the ligands to the site in mitochondria may be controlled by endogenous phospholipids.

Let us discuss some of the implications and future prospects. One of the promising directions of our work would be to study the relationship between the effect of inhibition of CcO by steroid hormones and the tissue specificity of the enzyme. The first stage of steroidogenesis proceeds in mitochondria of adrenal glands, gonads, and some other organs. This is the conversion of cholesterol into pregnenolone which is catalyzed by the cytochrome P450 side chain cleavage enzyme system located in the inner mitochondrial membrane, see Stocco [35]. Apparently, CcO in steroidogenic tissues must function in the presence of at least these two steroid compounds, and their inhibitory action in this case looks unlikely. According to our unpublished data, cholesterol actually does not inhibit the oxidase activity of the bovine heart enzyme. It is of interest to test pregnenolone from this point of view. Additionally, it would be useful to test the sensitivity of CcO from steroidogenic tissue to estradiol and testosterone.

On the other hand, the presented study was performed on enzyme from the heart, which is one of the target organs for steroid hormones. Cytoplasmic and nuclear receptors for sex hormones and some other steroids including vitamin D were found in cardiomyocytes long ago, see Stumpf et al. [36]. The physiological effect of “cardiac glycosides” is probably based just on their structural similarity with steroids. The effects of steroid hormones on the tissue of heart and blood vessels are diverse and very complex—this is the subject of studies by physiologists and of active discussion in the special medical literature. Without being clinicians, we only venture to suggest that the inhibition of the respiratory chain at the CcO level by steroid hormones may play a prominent role in the whole picture which should be clarified.

Mitochondrion, in its turn, is a target organelle for steroid hormones. All currently known cases of steroid interaction with mitochondria are receptor-dependent. They include transcriptional regulation of nuclear- as well as mitochondrial-encoded mitochondrial proteins, and the effects on mitochondria due to interactions with cytoplasmic signaling peptides and non-genomic control of cation fluxes (reviewed by Gavrilova-Jordan and Price [37]). From a global point of view, it seems very remarkable that the same regulatory molecule (steroid hormone) implements its influence on metabolism in two parallel ways: indirectly, through complex cascades of reactions, and directly, by selective interaction with the key respiratory enzyme resulting in modulation of its activity. Presumably, such double regulation makes the system more flexible: for example, it could accelerate the response to a stimulus, increase the control over responses, etc. We hope that our work will stimulate further research on this problem.

## Figures and Tables

**Figure 1 cells-09-02211-f001:**
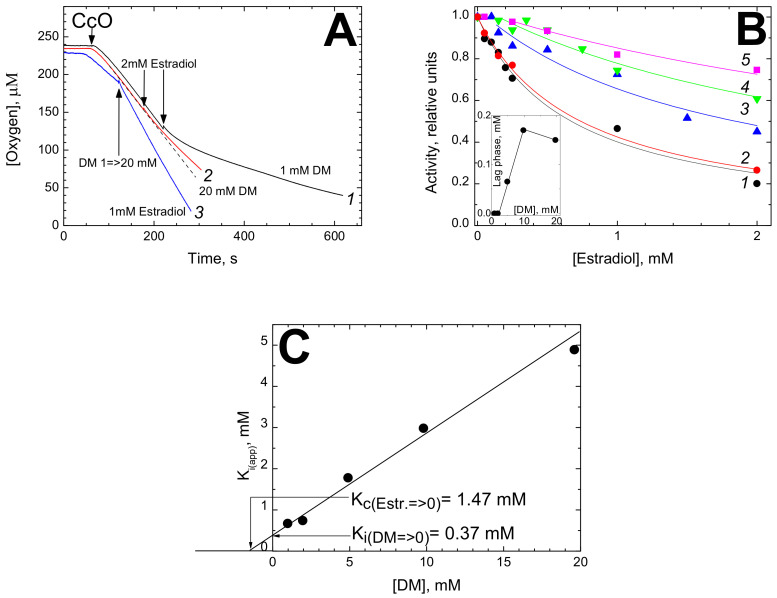
Inhibition of cytochrome *c* oxidase activity by estradiol. (**A**) Oxygen consumption was registered in the presence of 1 mM (traces *1*, black and *3,* blue), or 20 mM (trace *2*, red) dodecyl-maltoside (DM). The reaction was triggered by addition of 24 nM CcO. Trace *3*—the medium contained initially 1 mM estradiol. Other additions are indicated by the arrows. The traces have been corrected for ascorbate autoxidation. Dashed line overlapping trace *2* shows the kinetics in the absence of estradiol. (**B**) Titration of CcO activity by estradiol at different concentrations of DM. The experimental conditions, essentially as above. DM concentration: *1* (black circles)—1 mM; *2* (red circles)—2 mM; *3* (blue triangles)—5 mM; *4* (green triangles)—10 mM; *5* (magenta squares)—20 mM. The reaction was initiated by addition of CcO, and estradiol was added in 5 min after the onset of respiration. The ordinate axis represents the respiration rate after estradiol addition normalized to the initial rate. Theoretical curves are drawn through the experimental points starting from the end of the lag-phase (see the text). Inset shows dependence of the lag-phase length on DM concentration (filled circles). (**C**) Dependence of an apparent K_i_ for estradiol on DM concentration. The segment being cut off on the Y axis indicates the true K_i_ value in the absence of DM, the segment being cut off on the X axis in its negative area indicates the value of dissociation constant for DM in the absence of estradiol (Estr.), K_c_ (both segments are pointed out by arrows).

**Figure 2 cells-09-02211-f002:**
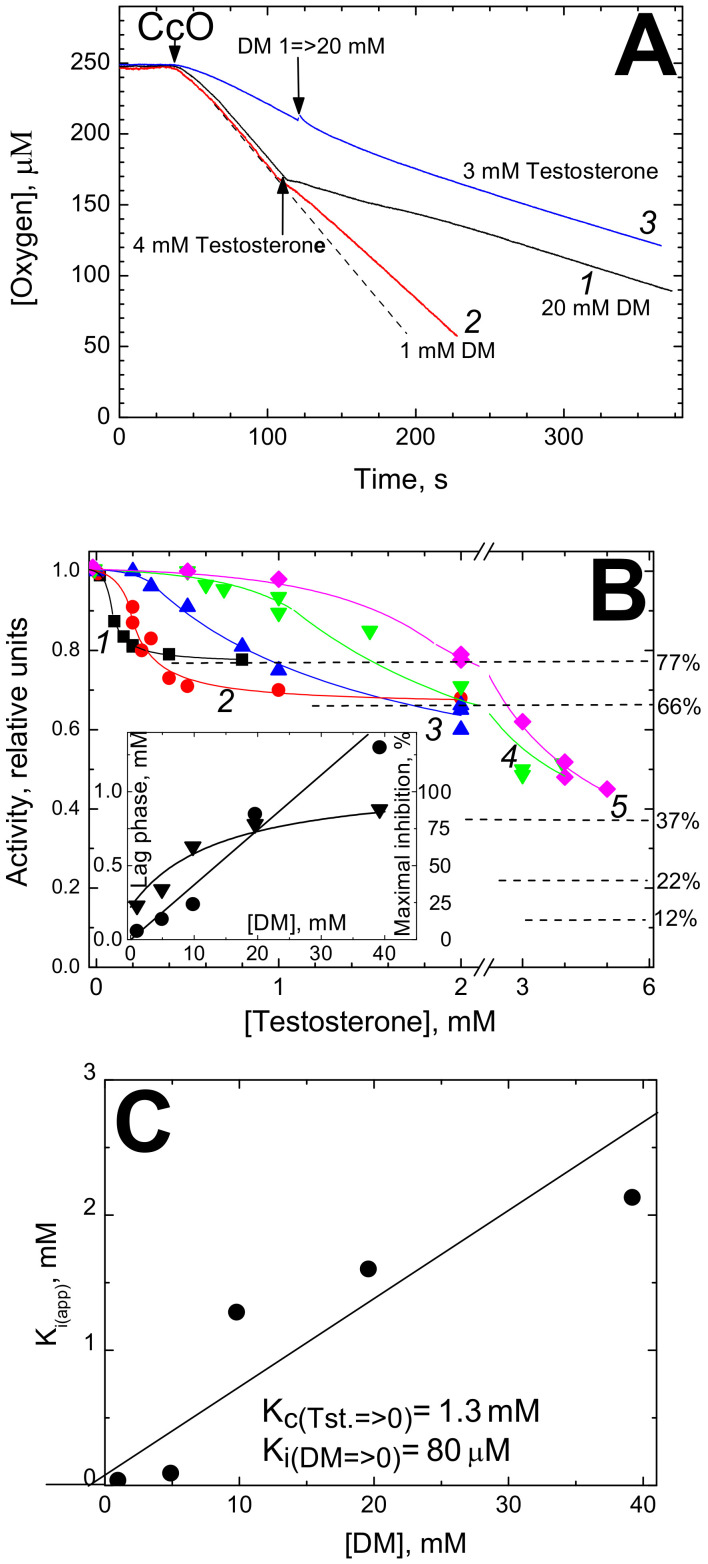
Inhibition of cytochrome *c* oxidase activity by testosterone. (**A**) Oxygraph traces. Basic conditions as in Figure 1A. The initial DM concentrations are: trace *1* (black)—20 mM, traces *2* (red) and *3* (blue)—1 mM. Trace *3*—the medium was also supplemented with 3 mM testosterone. Other additions are indicated by the arrows. The dashed straight line is drawn to visualize the effect of testosterone at 1 mM DM. (**B**) Titrations of CcO activity by testosterone at different concentrations of DM. The DM concentrations were: *1* (black squares)—1 mM; *2* (red circles)—5 mM; *3* (blue triangles)—10 mM; *4* (green triangles)—20 mM; *5* (magenta diamonds)—40 mM. Other conditions are mostly as in Figure 1B. Theoretical curves described by hyperbolic function (see the text) are drawn through the experimental points in the range of testosterone concentrations above the lag-phase value. For the lower concentrations of the inhibitor, the experimental points are connected by empirical lines simply to guide the eye. Dependences of the lag-phase length (filled circles, the left-hand Y axis) and of maximal inhibition level (filled triangles, the right-hand Y axis) on DM concentration are shown in the Inset. (**C**) Dependence of an apparent K_i_ for testosterone on DM concentration. The true value of K_i_ for testosterone (Tst.) in the absence of DM and K_c_ (dissociation constant) for DM in the absence of testosterone can be graphically determined as in the case of estradiol (see Figure 1C).

**Figure 3 cells-09-02211-f003:**
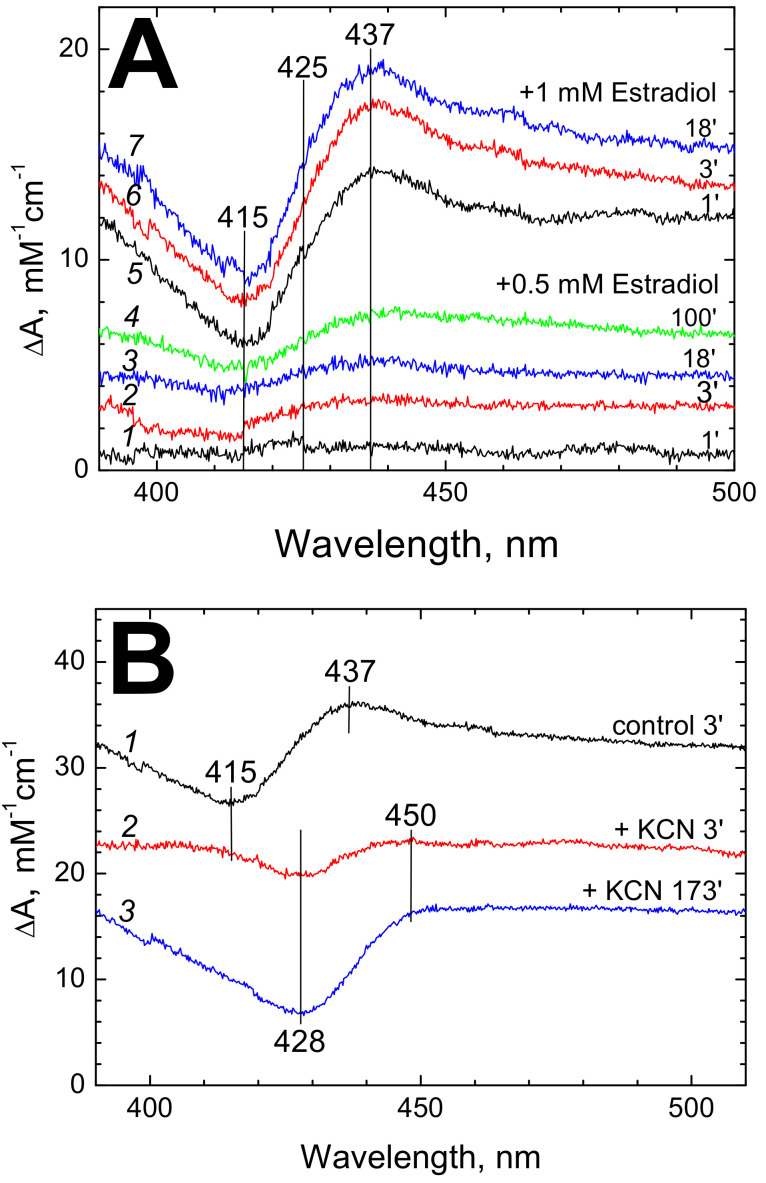
Estradiol-induced spectral changes of CcO. (**A**) Difference spectra induced by addition of 0.5 mM (*1–4*) or 1 mM (*5–7*) estradiol benzoate to the air oxidized CcO (1.22 µM). Spectra were recorded at the indicated time intervals after the hormone addition. The basic medium (pH 8.0) was supplemented with 13 mM DM. (**B**) Effect of cyanide on the estradiol-induced spectral shift of heme *a_3_*^3+^. Spectrum *1* (black)—control (difference spectrum recorded 3 min after addition of 1 mM estradiol to the air oxidized CcO). Spectrum *2* (red) was obtained as *1* but with the cyanide-complexed CcO (produced by 1 h incubation of CcO in the assay medium supplemented with 5 mM KCN and 25 µM potassium ferricyanide which prevented oxidized cyanide adduct from slow reduction). Spectrum *3* (blue)—as *2* but recorded 3 h after estradiol addition. Other conditions are as in panel A. (**C**) Time dependence of the estradiol-induced spectral changes. Traces *1* (black circles) and *2* (red triangles) represent time evolution of absorption difference at (437–415 nm) caused in the ligand-free oxidized CcO by incubation with 0.5 mM and 1 mM estradiol, respectively (see panel A). Trace *3* (blue triangles)—development of absorption difference at (428–450 nm) induced in the cyanide-complexed CcO by incubation with 1 mM estradiol (see panel B).

**Figure 4 cells-09-02211-f004:**
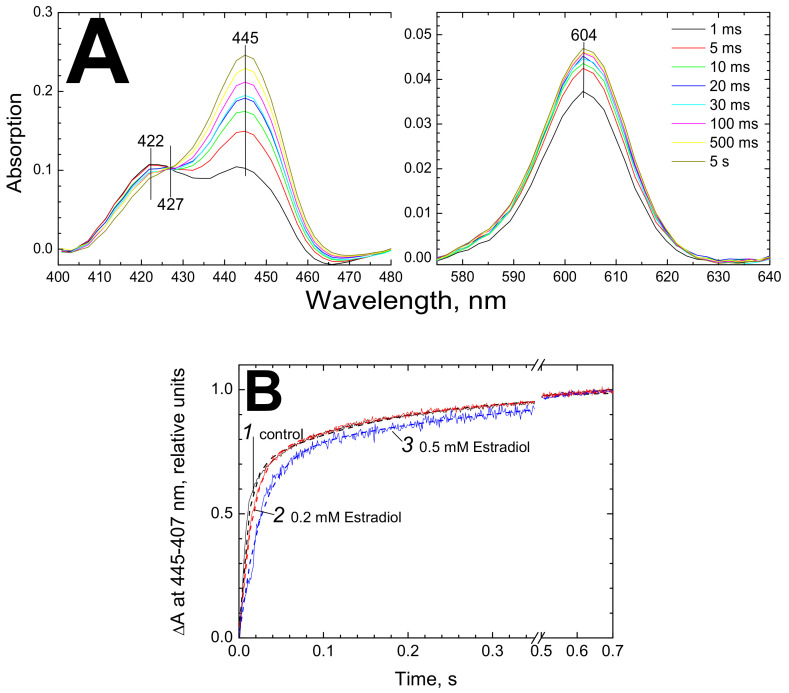
Estradiol inhibits reduction of heme *a_3_*. (**A**) Series of absolute spectra obtained upon rapid mixing of CcO with strong reductant using diode array spectrophotometer SX-20. Final concentrations: 3 µM CcO, 20 mM sodium dithionite, 5 mM RuAm. The left and right panels represent the Soret and the visible regions of the spectra, respectively. (**B**) Kinetics of absorption difference at (445 nm–407 nm) upon rapid reduction of the air-oxidized CcO is shown (see panel A for experimental conditions). Trace *1* (black)—control, trace *2* (red)—CcO was treated with 0.4 mM estradiol before mixing with the reductant, *3* (blue)—CcO was treated with 1 mM estradiol before the mixing. The experimental data are superimposed on the theoretical curves (dashed traces) described by the function: f(x) = A_1_·(1-exp(-x/τ_1_))+ A_2_· (1-exp(-x/τ_2_)), where A_1_, A_2_ are normalized amplitudes and τ_1_, τ_2_ are characteristic times of the rapid and slow phases, respectively (see the text).

**Figure 5 cells-09-02211-f005:**
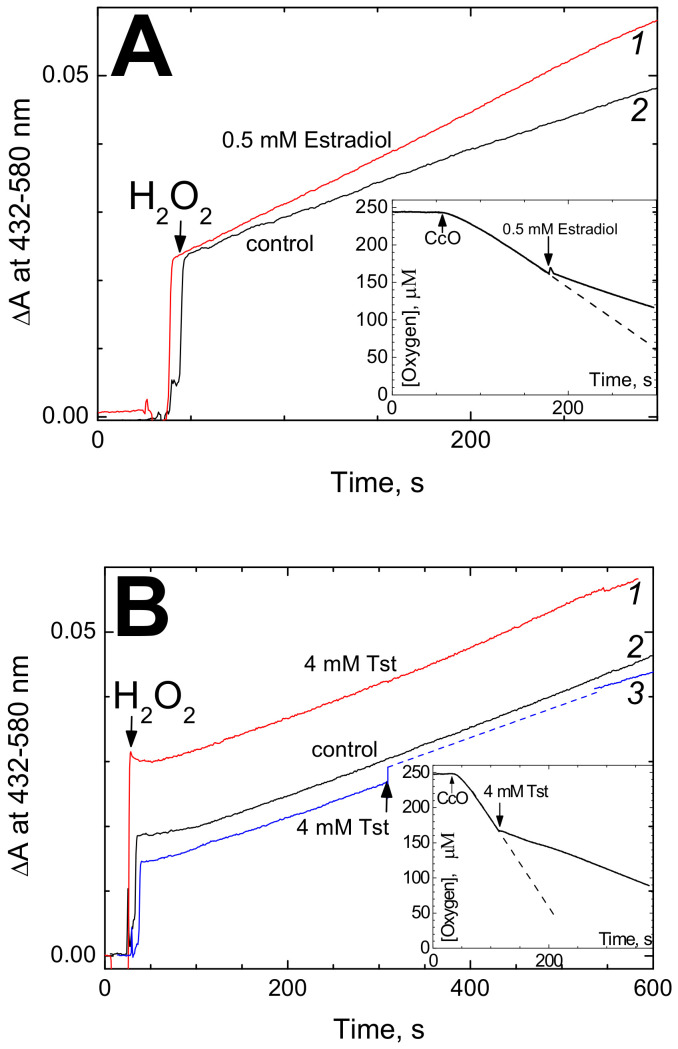
Effect of steroid hormones on the peroxidase activity of CcO. (**A**) Effect of estradiol. Peroxidation of *o*-dianisidine was monitored by increase of absorption at 432 nm vs. 580 nm as a reference. The basic reaction medium (pH 7.6) with 1 mM DM contained 0.2 mM *o*-dianisidine and 0.6 µM CcO. The reaction was initiated by addition of 4 mM H_2_O_2_. Trace *1* (red)—recording in the presence of 0.5 mM estradiol benzoate, trace *2* (black)—the control. The initial jump upon H_2_O_2_ addition corresponds to spectral response of heme *a_3_* in CcO. Inset: 0.5 mM estradiol in the presence of 1 mM DM does inhibit cytochrome oxidase activity of CcO (see Figure 1A for more details). (**B**) Effect of testosterone (Tst). The conditions are as in panel A except that concentration of DM in the buffer was 20 mM. Trace *1* (red)—recording in the presence of 4 mM testosterone, trace *2* (black)—the control, trace *3* (blue)—testosterone (4 mM) was added during the reaction time course (indicated by the arrow), which was followed by temporary increase of turbidity (the supposed trajectory until spectral registration became possible is shown by the dashed line). Inset: 4 mM testosterone in the presence of 20 mM DM does inhibit oxidase activity of CcO (see Figure 2A for other conditions).

**Table 1 cells-09-02211-t001:** Steroid regulatory molecules affecting CcO.

	Inhibition of CcO Oxidase Activity		Inhibition of CcO Peroxidase Activity	Spectral Response Induced in Oxidized CcO
	DM Influence	K_i_, mM (DM→0)		
**Estradiol**	competes 1:1	0.37	no	red shift of *a_3_*^3+^; ε = 10 mM^−1^⋅cm^−1^
**Testosterone**	competes 1:1; increases maximal level of inhibition	0.08	no	no
**Dehydroepiandrosterone (DHEA)**	competes 1:1	0.11	no	no data
**Progesterone**	competes 1:1	0.12	no	no data
**Cholecalciferol** **(vitamine D3)**	competes 1:1	0.009	yes (K_i_ ~0.2 mM)	red shift of *a_3_*^3+^; ε = 25 mM^−1^⋅cm^−1^
**Ergocalciferol** **(vitamine D2)**	competes 1:1	0.02	no data	no data

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
