# Peer review of "Interaction of Cytochrome C Oxidase with Steroid Hormones"

_cells, 2020, doi:10.3390/cells9102211_

Round 1

Reviewer 1 Report

The manuscript entitled “Interaction of cytochrome c oxidase with steroid hormones” by Oleynikov I, Azarkina N, Vygodina T and Konstantinov A is a continuation of research over mechanism of cytochrome c oxidase inhibition. The authors extend their research on interaction of cytochrome c oxidase with steroid hormones.

Suggestion:

Discussion

Since cytochrome c oxidase was purified from bovine heart, physiological significance of studied CcO inhibition by steroid hormones is important. Hence, it would be of great interest to readers to discuss the possible effects of deficiency or excess of estrogen and testosterone on the hearts CcO.

Also, could locally produced steroid hormones by mitochondria, be able to modulate mitochondrial activities via modulation of CcO? Potential significance?

Abstract

Line 11: Introduce abbreviation for cytochrome c oxidase (CcO)

Introduction

Line 34: close Parentheses

Author Response

We are grateful to the Reviewer 1 for the valuable comments. Our reply is below.

 Line 11: Introduce abbreviation for cytochrome c oxidase (CcO)

  • Done

Line 34: close Parentheses

  • Corrected

Comment:

Since cytochrome c oxidase was purified from bovine heart, physiological significance of studied CcO inhibition by steroid hormones is important. Hence, it would be of great interest to readers to discuss the possible effects of deficiency or excess of estrogen and testosterone on the hearts CcO.

Reply:

The heart is one of the «target» organs for steroid hormones. Cytoplasmic and nuclear receptors for sex and some other steroids including vitamin D have been found in cardiomyocytes long ago. The physiological effect of "cardiac glycosides" is probably based just on their structural similarity with steroids. However the effects of steroid hormones on the tissue of heart and blood vessels are diverse and very complex – this is the subject of studies by physiologists and of active discussion in the special medical literature. Considering the complexity of the whole system we do not undertake to predict what particular consequences for the heart will result from excess or deficiency of sex hormones. However, the coexistence in the same tissue of two parallel ways of implementing the regulatory function of a hormone (canonical, through multilevel cascades of reactions, and direct, based on the interaction with a key respiratory enzyme and modulation of its activity) looks quite remarkable.  It seems like a new precedent in the field of regulation.

The correspondent remark is added to the revised version of our paper (see lines 458-466).

Comment:

Also, could locally produced steroid hormones by mitochondria, be able to modulate mitochondrial activities via modulation of CcO? Potential significance?

 Reply:

Indeed, the first stage of steroidogenesis (the conversion of cholesterol into pregnenolone) is catalyzed by the cytochrome P450 side chain cleavage enzyme system located in the inner mitochondrial membrane. Thus, CcO in steroidogenic tissues is inevitably subject to constant exposure to at least these two steroid compounds. However, we would expect no inhibition in this case. According to our unpublished data, cholesterol actually does not inhibit the oxidase activity of the bovine heart enzyme. It is interesting to test pregnenolone from this point of view. As well, it is interesting to test the sensitivity of CcO from steroidogenic tissue to estradiol and testosterone.

The correspondent remark is added to the revised version of our paper (see lines 448-457).

Reviewer 2 Report

First of all, I would like to thank the invitation to review this paper. Although it not fall completely in my area of expertise, I have to congratulate the work performed. This is an original, well-written manuscript, with good scientific soundness, novelty, and innovation, besides to contribute to the comprehension of some effects triggered by hormones.

I have some minor comments to raise:

  • abstract should be completely restuctured and organized according to the journal guidelines (i.e., background, methods, results and conclusions)
  • l. 46-52: restructure this paragraph, first highlighting the data obtained so far and then define the question that motivate the realization of this work
  • whole manuscript: instead of using "ref. [X], please refer to the name of authors who carried out the study
  • discussion and conclusions sections: from the findings obtained here, what can authors speculate to clinical practice? will these advances trigger changes in clinical interventions? what can authors infer on the in vivo findings from the data obtained here?

Author Response

We are grateful to the Reviewer 2 for the valuable comments. Our reply is below.

 Comment:

abstract should be completely restuctured and organized according to the journal guidelines (i.e., background, methods, results and conclusions)

Reply:

We would not like to change the structure of the abstract, since in its current form, in our opinion, it introduces the reader to the essence of the study most comprehensively and briefly. Though we can do this at the insistence of the editors.

Comment:

  1. 46-52: restructure this paragraph, first highlighting the data obtained so far and then define the question that motivate the realization of this work

Reply:

This paragraph (lines 47-54 in the revised version of the manuscript) is a direct continuation of the previous one (lines 35-46), which says exactly what you are writing about: we start with the background (the results of Ferguson-Miller group: the BABS site is descrìbed in the crystal structure of CcO from R. sphaeroides and mitochondria; structural considerations rank steroid hormones high in the list of potential ligands of the BABS), then the motivating question follows (no effect of steroid hormones on the activity of CcO from R. sphaeroides was found, despite the expectations). After this, the next paragraph (lines 47-54) begins with the reasons for choosing the object of research (we suggested that CcO from animal mitochondria may be a more appropriate object for the studies of hormone action than the bacterial enzyme) following by a summary of the main points of the presented study.

Comment:

whole manuscript: instead of using "ref. [X], please refer to the name of authors who carried out the study

Reply:

Done in the cases of the most important references.

Comment:

discussion and conclusions sections: from the findings obtained here, what can authors speculate to clinical practice? will these advances trigger changes in clinical interventions? what can authors infer on the in vivo findings from the data obtained here?

Reply:

Without being clinicians, we can only speculate on the use of our results in medicine. Nevertheless, we would suggest taking into account the possibility of direct inhibition of the respiratory chain at the CcO level when prescribing steroid drugs. Artificial analogs of steroid hormones are widely used: as anti-inflammatory drugs, in hormone replacement therapy, etc. Anabolic-Androgenic steroids (AAS) are used in sports and for bodybuilding, which is often followed by undesirable consequences from the cardiovascular system. The possible effect of such drugs on the intracellular energy balance has never been taken into account. Information on the ability of some steroids to inhibit mitochondrial CcO may be useful in the development of new drugs and to prevent side effects from their use.

We added to the text a short remark on this subject (see lines 462-466). We cannot go into details since this problem is beyond the scope of our paper.

Reviewer 3 Report

The study by Oleynikov and colleagues is an elegant work on the inhibition of cytochrome c oxidase by steroid hormones. I find the data interesting and sound, although I am not convinced of the linearity of the Ki(app) vs DM relationship (figure 2C). The authors should clarify this point.

The discussion is very detailed and complete, although I suggest the authors to add some thoughts on the reasons why cytochrome c oxidase should be inhibited in vivo by seroid hormones.

Author Response

We are grateful to the Reviewer 3 for the valuable comments. Our reply is below.

 Comment:

I am not convinced of the linearity of the Ki(app) vs [DM] relationship (figure 2C). The authors should clarify this point.

Reply:

We agree with your comment regarding Fig. 2C. Unfortunately, titration of oxidase activity with testosterone occurred to be a technically hard task due to the low solubility of testosterone, which led to an increase in measurement errors. However, we believe that the dependence of Ki(app) on [DM] in this case is linear indeed, since it was linear in all other cases with other steroid compounds tested (see Table 1).

The correspondent remark is added to the revised version of our paper (see lines 201-203).

Comment:

The discussion is very detailed and complete, although I suggest the authors to add some thoughts on the reasons why cytochrome c oxidase should be inhibited in vivo by steroid hormones.

Reply:

We can only speculate on this topic. Notably, steroids regulate the synthesis of enzymes of oxidative phosphorylation at the transcriptional level and through other regulatory elements. It seems very remarkable that the same molecule (steroid hormone) implements its influence on oxidative phosphorylation in two parallel ways: indirectly, through complex cascades of reactions, and by direct interaction with the key respiratory enzyme, with modulation of its activity. From this point of view, the interaction of steroids with CcO looks as an important biological precedent. Perhaps, such a double regulation could provide a faster response of the system to the stimulus (the appearance of a hormone). It is also possible that a negative feedback could arise between the direct and indirect action of the hormone, which attenuated the induced effects. For example, inhibition of the respiratory chain at the level of CcO reduces the energy balance of the cell, which results in slowing down of all energy-dependent reactions triggered by the canonical action of the hormone.

The correspondent remark is added to the revised version of our paper (see lines 467-476).